# Monitoring by a Sensitive Liquid-Based Sampling Strategy Reveals a Considerable Reduction of *Listeria monocytogenes* in Smeared Cheese Production over 10 Years of Testing in Austria

**DOI:** 10.3390/foods10091977

**Published:** 2021-08-24

**Authors:** Peter Zangerl, Dagmar Schoder, Frieda Eliskases-Lechner, Abdoulla Zangana, Elisabeth Frohner, Beatrix Stessl, Martin Wagner

**Affiliations:** 1Higher Federal Teaching and Research Institute in Tyrol for Agriculture and Nutrition as well as Food and Biotechnology, Rotholz 50, 6200 Strass im Zillertal, Austria; peter.zangerl@hblfa-tirol.at (P.Z.); frieda.eliskases-lechner@hblfa-tirol.at (F.E.-L.); 2Unit of Food Microbiology, Institute of Food Safety, Food Technology and Veterinary Public Health, Department for Farm Animals and Veterinary Public Health, University of Veterinary Medicine Vienna, Veterinärplatz 1, 1210 Vienna, Austria; dagmar.schoder@vetmeduni.ac.at (D.S.); Abdoulla.Zangana@vetmeduni.ac.at (A.Z.); Elisabeth.Frohner@vetmeduni.ac.at (E.F.); Beatrix.Stessl@vetmeduni.ac.at (B.S.); 3Austrian Competence Center for Feed and Food Quality, Safety and Innovation (FFOQSI), Technopark C, 3430 Tulln, Austria

**Keywords:** *Listeria* spp., *Listeria monocytogenes*, prevalence, detection, monitoring, smear

## Abstract

Most Austrian dairies and cheese manufacturers participated in a *Listeria* monitoring program, which was established after the first reports of dairy product-associated listeriosis outbreaks more than thirty years ago. Within the *Listeria* monitoring program, up to 800 mL of product-associated liquids such as cheese smear or brine are processed in a semi-quantitative approach to increase epidemiological sensitivity. A sampling strategy within cheese production, which detects environmental contamination before it results in problematic food contamination, has benefits for food safety management. The liquid-based sampling strategy was implemented by both industrial cheese makers and small-scale dairies located in the mountainous region of Western Austria. This report considers more than 12,000 *Listeria* spp. examinations of liquid-based samples in the 2009 to 2018 timeframe. Overall, the occurrence of L. monocytogenes in smear liquid samples was 1.29% and 1.55% (*n* = 5043 and *n* = 7194 tested samples) for small and industrial cheese enterprises, respectively. The liquid-based sampling strategy for Listeria monitoring at the plant level appears to be superior to solid surface monitoring. Cheese smear liquids seem to have good utility as an index of the contamination of cheese up to that point in production. A modelling or validation process should be performed for the new semi-quantitative approach to estimate the true impact of the method in terms of reducing Listeria contamination at the cheese plant level.

## 1. Introduction

Cheese products have been a possible source of outbreaks of listeriosis for many decades, especially smeared cheeses and those made from raw milk [1,2,3] (https://www.cdc.gov/Listeria/outbreaks/index.html; accessed on: 19 June 2021).

Cheeses made from goat or sheep milk are particularly likely to be *L. monocytogenes* positive (3.6–12.8%) [4]. This is also evident from a search of the portal for Food and Feed Safety Alerts (RASSF), where 39/90 *L. monocytogenes* notifications relate to cheeses made from goat or sheep milk (https://webgate.ec.europa.eu/rasff-window/screen/search; accessed on: 19 June 2021). Significant genetic diversity was identified among *L. monocytogenes* strains through the use of molecular epidemiology methods [5,6,7,8,9,10]. Other research groups noticed an increased occurrence of hypervirulent *L. monocytogenes* strains of genetic lineage I (serovar 1/2b, 4b, sequence type (ST)1, ST4, ST6) in the dairy niche [11,12]. In addition, *L. monocytogenes* genetic lineage II strains (e.g., ST7, ST14, ST204; ST451), including hypovirulent types (ST121, ST9) were reported to persist in the dairy processing environment, potentially due to the intra- and inter-species exchange of mobile genetic elements [6,13,14,15,16,17,18]. 

An important role in environmental adaptation is played by highly conserved plasmids circulating worldwide in a distinctive *L. monocytogenes* gene pool [9,19,20,21]. These more complex epidemiological considerations have a direct impact on surveillance used to verify the effectiveness of *L. monocytogenes* controls within food safety management systems.

Although milk is usually subjected to a heating process prior to processing, cheese can become contaminated during several process steps such as pressing, curing, ripening, and during cutting and packaging [22,23]. 

In food processing environments (FPEs), contamination is often related to *L. monocytogenes’* colonization of surfaces, including in the dairy sector [24].

Own-check systems are applied with a focus on testing end products and samples from the production environment according to EC regulation 2073/2005 [25]. In food processing environments (FPEs), contamination is often related to *L. monocytogenes’* colonization of surfaces, including in the dairy sector [25].

In particular, newly built manufacturing plants or plants undergoing reconstruction measures are at high risk of being colonized with *L. monocytogenes* [26,27]. 

In cases where *L. monocytogenes* is detected on the end product at unacceptable levels, withdrawals from the market or recalls are implemented to protect the safety of the consumer. 

To minimize the risk of process contamination during cheese ripening via the cheese smear, this liquid-based sampling strategy was established, which is also applicable to brine or drain water samples [28] (Figure 1). Since the majority of soft, semi-hard and hard cheeses in Austria are surface-ripened, smear liquids are, in most cases, collected after the smearing process. Compared to product-contact surface-sampling using friction-swabs, these liquids constitute a matrix that provides a much broader representation of the contamination status by including both cheese components and contact with surfaces inside of the production equipment, e.g., smear robots [29]. Sampling of a non-homogenous solid product creates real challenges in terms of consistency and representativeness. *Listeria* contamination is more likely on the surface rind than inside the cheese matrix. Moreover, sampling of a batch of individual cheeses has potential for statistical biases unless true randomisation is rigorously adhered to [3,30]. Sampling biases are major concerns and the degree of harmonization among procedures is usually low (sampling frequency and sampling sites are usually less well standardized) [31]. The implementation of preventive food safety concepts by tailored food sector-specific sampling procedures provokes a deepened insight of the FBOs into the operation-specific status of contamination and facilitates a comparison of scenarios.

The monitoring of cheeses produced without smearing focuses on sampling liquids including brine, wash water (water used to clean production devices such as trolleys or trays) or drain water. Sampling events depend on ripening time and batch size and should be performed twice per month. For small-scale dairies, the sampling frequency should ensure that every cheese is included at least once during ripening. After detection of *L. monocytogenes* and *Listeria* spp. by ISO enrichment methods, PCR-based species differentiation should be performed on typical *Listeria* colonies isolated on selective agar [32,33]. Persistence of *L. innocua* was shown to occur more frequently than persistence of *L. monocytogenes* and is, therefore, seen as an indicator of inadequate hygiene [34,35]. 

If *L. monocytogenes* is detected, rigorous sanitation of the facility is essential. Additionally, the sample number is increased and testing entails end products and further environmental samples (e.g., tanks, racks, conveyor belts and ventilation). This step includes a microbiological investigation post sanitation to verify the efficiency of the measures taken. If desired, a facility inspection audits the internal traffic management and checks other elements of the prerequisite programs (PrPs) that are in place, such as the maintenance of buildings and rooms. The hygienic status of production is, therefore, checked stepwise at all production areas. At the heart of the monitoring and surveillance approach is the range of sample volume that is tested: 600 to 800 mL (two labs involved, method slightly deviates), 100 mL, 10 mL, and 1 mL of liquid (Figure 1). This semi-quantitative way of testing both low and high sample volumes substantially increases the epidemiological sensitivity of the method due to a higher quantity of sample matrix. 

Indeed, directly after initial contamination of either the environment or the food, *L. monocytogenes* might be scarcely detectable in food business operations (FBOs) and testing of high volumes increases the likelihood of finding low contamination levels. 

Therefore, the aim of this study was to present the alternative semi-quantitative liquid-based sampling strategy to increase the epidemiological sensitivity in the detection of *L. monocytogenes* and other *Listeria* species. For this purpose, the alternative method was implemented within the framework of *Listeria* monitoring, for both industrial cheese makers and small-scale dairies located in the mountainous region of Western Austria. By using this approach, more than 12,000 samples were tested during the period from 2009 to 2018. 

## 2. Materials and Methods

### 2.1. Materials

Testing of cheeses for *L. monocytogenes* with a high level of confidence is limited by statistical biases. Investigation of smear liquid samples for monitoring purposes is a highly informative sampling strategy as all cheeses of a lot are usually treated with a smear liquid from the same tank. Therefore, analysis of the smear liquid allows for the contamination status of the entire cheese lot being stored for ripening. Sampling of smears is relatively simple and no cheese is damaged or spoiled by the sampling procedure [36] (Sampling scheme Figure 1). 

### 2.2. Companies

According to the Austrian trade register for companies, around 80 professional cheese producers (this number does not include farm dairies directly marketing the product) exist in Austria (https://www.firmenbuchgrundbuch.at/ accessed on: 19 June 2021). Cheese making in Austria is conducted in operations that vary in size, ranging from small (products merchandised regionally) to industrial (products mostly merchandised across all of Austria and export markets such as the EU-27). Whereas some companies process a couple thousand liters of milk per year, industrial companies (spread over entire Austria) process tens of millions of liters. Small-scale cheese makers are mostly located in the Western parts of Austria. Many of them send their samples to the Higher Federal Teaching and Research Institute Tyrol (HBLFA) and, depending on the year, between 51 and 75 companies participate (see Table 1). The number of large industrial cheese producers that cooperate with the Institute of Food Safety, Food Technology and Veterinary Public Health (IFFV) ranges from 7 to 9, and these companies produce more than 80% of the industrially produced smeared soft and semi-hard cheeses in Austria. 

### 2.3. Methods

A total of 12,237 smear liquid samples were examined in the years 2009–2018 (see Table 1) by both testing labs. Liquid smear samples were collected in two-month intervals from industrial cheesemakers. Small FBOs collected smear samples during cheese ripening, representing comparatively smaller batches. Sample volumes of 1 mL (IFFV only), 10 mL, 100 mL and 600 mL (IFFV) or 800 mL (HBLFA) are routinely investigated. The occurrence of *L. monocytogenes* in products, product-associated samples and in the processing environment is considered to be rather low and not equally distributed; therefore, the semi-quantitative enrichment protocol is assumed to increase the detection of *L. monocytogenes* in at least one of the enrichment steps [28]. One liter of liquid sample was divided into 4 preparations as follows: 600 or 800 mL were centrifuged at 4800 rpm for 30 min at 4 °C (Beckman Coulter, Brea, CA, USA). The sediment was completely transferred into 1 L Half-Fraser broth (Biokar Diagnostics-Solabia Group, Pantin Cedex, France). Subsequent preparation steps of the semi-quantitative approach included 100 mL, 10 mL, and 1 mL diluted 1:10 in Half-Fraser broth (Biokar Diagnostics-Solabia Group).

Sample enrichment in Half-Fraser broth and Fraser broth (both Biokar Diagnostics-Solabia Group) and strain isolation on Palcam Agar (Biokar Diagnostics-Solabia Group) and *Listeria* agar acc. Ottaviani and Agosti (ALOA; Merck KGaA, Darmstadt, Germany) was performed according to ISO 11290:1 [33]. In detail, for each semi-quantitative enrichment scenario (i.e., 600/800 mL, 100 mL, 10 mL, 1 mL), following 24 h incubation at 30 °C in Half-Fraser broth, aliquots of 100 μL were transferred to 10 mL Fraser broth and then incubated for 48 h at 37 °C.

In addition, at IFFV, polymerase chain reaction (PCR) assays targeting the *hly* gene (encoding the pore-forming cytolysin listeriolysin) and *iap* (invasion-associated protein p60) gene [31,37] were included for species confirmation (for technical details, see Asperger et al. [28]). This approach ensured that even a single *L. monocytogenes* colony that may have hidden in a plethora of other microorganisms, such as *Bacillus* spp. growing on PALCAM or chromogenic agar, would be detected [38]. 

The DNA extraction was performed directly from selective agar plates by rinsing the surface with 1 mL of 0.01 M Tris HCl buffer (Sigma Aldrich Corp., St. Louis, MO, USA). The suspension was centrifuged for 5 min at 8000 rpm and the pellet was suspended in 100 μL 0.01 M Tris HCl Buffer (Sigma Aldrich Corp.) and vortexed. In parallel, material from *L. monocytogenes* subcultures (1–2 colonies) was suspended in 100 μL Tris HCl Buffer. Subsequently, 400 μL Chelex^®^ 100-Resin (BioRad, Hercules, CA, USA) was added to the bacterial suspension, heated for 10 min at 100 °C and centrifuged at 14,000 rpm for 5 s [39]. The DNA supernatant was transferred to Maxymum Recovery tubes (VWR International-Avantor, Radnor, PA, USA) and stored at −20 °C before downstream processing [31,37]. The PCR-amplicons were electrophoretically separated in a 1.5% agarose gel containing 0.5× Tris-Borate-EDTA (TBE) buffer and 3.5 μL peqGREEN DNA gel stain (VWR International-Avantor), at 120 V for 30 min. The DNA standard Thermo Scientific™ GeneRuler™ 100 bp (Thermo Fisher Scientific Inc., Waltham, MA, USA) was applied for fragment length comparison. The electrophoresis gels were photographed under UV light exposure (GelDoc 2000, BioRad, Hercules) and saved in tiff format for further comparison. 

## 3. Results and Discussion

*Listeria* contamination is an adverse event for many food business operations (FBOs), and the entire dairy sector suffers whenever outbreaks occur. A survey of technical managers in food processing plants on *L. monocytogenes* risk outcomes by Evans et al. [40]. revealed interesting assessments. Participants perceived a medium risk (on a scale from 1 to 10; 5.5) of *Listeria* in their operations with a high level of control and a high level of responsibility. In this study, technical leaders expressed concern regarding *L. monocytogenes* and indicated that increased awareness of the pathogen would improve control actions. Installing *Listeria* environmental monitoring was considered essential in this regard [40].

A recent evaluation of monitoring approaches by Magdovitz et al. [41] showed that facilities prefer to test environmental monitoring zones 2 through 4 (non-food contact areas). Few facilities actively integrate raw material controls and intermediate products or product-associated samples into their sampling plan [41].

Many data are available for *Listeria* contamination scenarios in single FBOs, but little information is available for whole food production sectors such as smeared cheese manufacturing. EU baseline data on *L. monocytogenes* prevalence in cheese samples at the end of shelf-life showed a rate of 0.47%, with 0.06% of samples exceeding the level of 100 cfu/g [42].

The few studies that are focused on data across food producers and batches are somehow comparable to our data and are cited in the following paragraph. Data on liquid-based sampling concepts are not available from the literature. Barría et al. [43] studied 546 cheese and milk samples to establish a monitoring system in Chilean cheese factories. *L. monocytogenes* was identified in 19 cheeses (4.1%), with a prevalence similar to that reported in a Polish study (6.2% *L. monocytogenes*, 370 samples) [44]. In both studies, the monitoring system focused on cheese samples as no food contact surface (FCS) or non-food contact surface (NFCS) samples were included in the sampling plan. Another *Listeria* spp. pilot study in PDO Taleggio cheese processing revealed a mean prevalence of 23.1% *Listeria*-positive samples (*n* = 360 samples). The ripening and cutting equipment were identified as high-risk areas for *Listeria* contamination [45]. Other short-term monitoring datasets were published, with an overall *L. monocytogenes* prevalence of 4.6% in various food sectors [46]. A larger dataset based on pathogen monitoring in small cheese processing plants (4430 samples; 6.03% *Listeria* spp.) suggested running routine sampling plans for at least 6 months and then evaluating appropriate sampling sites inclusively for *Listeria* occurrence [34]. 

In general, cheese surfaces are more likely to be contaminated by *L. monocytogenes* than the internal areas of the cheese. This was also the outcome of a baseline study, conducted at a national level, where Gorgonzola and Taleggio were the most frequently contaminated cheeses. Transmission of *L. monocytogenes* from contaminated cheese rind to the cheese interior during cutting or packaging is possible [47]. Therefore, product-associated samples, such as smear liquids and surface scrapings, should be considered in a *Listeria* monitoring program.

Our data from the cheese smear liquid-based monitoring showed, in small cheese producers (mainly soft and semi-soft cheeses), an average *Listeria* spp. (other than *L. monocytogenes*) and *L. monocytogenes* contamination of 15.6% and 4.5%, respectively, During the sampling period, an average of 67 out of 75 FBOs were *Listeria* spp. positive. Numbers for industrial cheesemakers show that an average of eight FBOs participated in the program, where means of 20.8% *L. monocytogenes* and 28.6% *Listeria* spp. (other than *L. monocytogenes*) were detected.

The *L. monocytogenes* contamination ranged from 0 to 12.5% and from 0 to 33.3% in small and industrial FBOs during 2009 to 2018, respectively. *Listeria* spp. other than *L. monocytogenes*, which were differentiated by the PCR approach [32], ranged from 8.1 to 20.6% in small FBOs and from 22.2 to 44.4% in industrial FBOs (Table 1), indicating that the latter was more highly contaminated with the potential pathogen. The industrial FBOs were higher contaminated with *L. monocytogenes* in comparison to small FBOs. Similar observations were made by Muhterem et al. [25], where the FPE of industrial cheesemakers indicated a higher *L. monocytogenes* contamination of up to 26% compared to farm cheesemakers (up to 6.4%). In total, *Listeria* spp. was detected in 4.19% (513 out of 12.237) of all smear liquid samples examined, whereas the percentage of *L. monocytogenes*-positive samples was 1.45% (178 out of 12.237 samples). The higher frequency of *Listeria* spp. (other than *L. monocytogenes*) contamination is an important indicator of necessary hygiene improvement measures to prevent *L. monocytogenes* from successfully establishing itself as a zoonotic pathogen in a FPE [48]. This value for *Listeria* spp.-associated contamination was substantially lower in comparison to samples that were tested at the IFFV between 1990 and 1999 (industrial cheese makers only: 14.09%) [28]. If calculated based on years, the prevalence of *L. monocytogenes* in smears was 0–4.4% (average: 1.29%) and 0–6% (average: 1.55%) for the small and the industrial cheese establishments, respectively (see Table 2). 

This is of interest as the industrial cheese producers included in this study mainly used pasteurized milk, while the small producers tended to use raw milk for the production of traditional specialty cheeses.

Since the occurrence of *L. monocytogenes* contamination was similar for both categories (Table 2), we confirmed that heat treatment of milk had little impact on the presence of *L. monocytogenes* in the smears and that, in the majority of our observations, cheese is more likely to become contaminated after coagulation [18,35,49]. 

Inclusion of high sample volumes was found to increase the detection sensitivity of the method as applied at both institutes. At HBLFA, 11.98% of samples tested positive in 800 mL and 100 mL but not in 10 mL, and 19.4% (*n* = 13) of all positive findings were found in the highest sample volume only (data not shown). Only 26.9% (*n* = 18) of all samples tested positive in 800 mL, 100 mL and 10 mL. From the fact that more than 30% of the positive events were observed in volumes of ≥100 mL only, we conclude that *L. monocytogenes* contamination levels are often very low at the beginning of a contamination event. Data also suggest that testing only 25 mL of cheese-associated fluids (which is commonly the case in other countries) does not provide enough epidemiologic sensitivity to detect low-level contamination. 

This assumption would be interesting to compare in the performance testing of the ISO method versus alternative liquid-based sampling strategies with higher sample volumes. Some samples revealed *L. monocytogenes* detection in either 10 mL or 100 mL but not in 800 mL. This effect could have been caused by a not-yet-understood antiListerial potential of the smear microbiota in some samples, testing too soon following the use of protective cultures against *L. monocytogenes* (e.g., phages), and extremely high numbers of accompanying flora after centrifugation of 600 and 800 mL, respectively [50,51,52]. Unpublished results on the inhibitory effects of smear samples on *Listeria* showed a highly variable pattern, ranging from a decrease in numbers of *L. monocytogenes* by 3 log units in some samples to a proliferation capacity of up to 4 log CFU/mL in other samples (Part, pers. communication). We conclude that testing of high volumes only is not sufficient to detect a contamination event; therefore, the more extensive approach of testing more than one sampling volume should be incorporated. Findings from small cheese producers were consistent with results that were found with samples originating from industrial cheese plants. Twenty-four percent (*n* = 22) of positive results were found in the high sample volume (600 mL) only. As many as 16.5% of smear liquid samples were found to be positive in sample volumes of 600 mL and 100 mL. Another 15.4% of the samples were positive in 600 mL, 100 mL and 10 mL. The smear monitoring conducted at IFFV also incorporated 1 mL samples. Being positive in 1 mL was thought to be a cause for concern as a higher number of *L. monocytogenes* might be present in the smear liquid and, subsequently, on the cheese. In 12% (*n* = 11) of all positive smear liquid samples, *L. monocytogenes* was found in all sample volumes (600 mL, 100 mL, 10 mL and 1 mL). As with the data provided by HBFLA, the findings at IFFV are inconclusive in some cases. In 24% of the positive results, *L. monocytogenes* was detected in 100 mL of sample volume only. 

Although the first food-associated outbreaks were reported from USA and Canada in the early 1980s, a game changer for the national dairy industry was the Swiss Vacherin Mont d‘Or outbreak in 1983–1987 [53]. Austrian companies began testing cheese brine and smears in 1988 to improve *L. monocytogenes* detection during production. From 1992 to 1994, a 30 to 40% positive test rate for *L. monocytogenes* was observed. Within a decade of increased measures, prevalence decreased to a detection rate of <5% [28]. The liquid-based sampling strategy also shows successful detection of *L. monocytogenes* in our approach, and possibly a positive impact in terms of avoiding false negatives and product withdrawals or recalls. This positive development of improved awareness of possible *L. monocytogenes* contamination occurred in spite of an ongoing restructuring of the dairy sector in Austria, which reduced the number of industrial cheese dairies from >50 in 1990 to less than 10 in 2019.

In line with the economic growth of some major players, the amount of produced cheese (soft, semisoft and hard cheese) quintupled from 1990 to 2019 (< 30,000 tons in 1995 to 131,000 tons per year in 2018).

The monitoring of results such as those achieved by the *Listeria* monitoring program is a prerequisite for the timely detection of potential safety hazards, including the contamination of cheese environments with *L. monocytogenes*. Frequent monitoring aids early *L. monocytogenes* detection, and prevents contamination and the placing of contaminated food on the market [31]. That there is a considerable likelihood for introduction is evidenced by the fact that, at least once, positive *Listeria* spp. results were revealed over all the years from a majority of the participants in the program. If contamination remains unaffected by routine hygiene measures, *Listeria* is spread within the production area through daily in-plant manipulations.

In the long run, *Listeria* spp. colonizes niches within the FBO, where the hygienic pressure is not high enough to prevent them from surviving, thereby allowing *Listeria* spp. to survive.

Experience in recent years has repeatedly confirmed that testing higher sample volumes effectively complements other hygiene inspection techniques, such as swabbing or contact sliding.

In accordance with the testing of product-associated liquids, environment-derived liquid samples such as drain water samples encompass the contamination status of large plant areas. The use of large volumes of liquid in our semi-quantitative sampling approach potentially reduces the false negative test results that can occur when using smaller volumes or simple contact sliding. 

Investigation of smear liquid is beneficial as this substrate is used on entire cheese batches for extended production periods. Therefore, with respect to cheese processing, the microbiological investigation of smear liquid is an appropriate parameter in any safety program dealing with smear-ripened cheeses.

Preventing foodborne hazards along the food processing chain is supported by an intelligent sampling strategy that may differ among food sectors and professionals. For *L. monocytogenes* environmental testing, mostly swab and sponge-based friction sampling methods are used [54]. The decrease in the *L. monocytogenes* detection rate, as seen in Austrian cheese factories in recent years, coincides with an increased understanding and acceptance of food safety parameters by the cheese producers, which was in part contributed to by a high-profile cheese-borne outbreak of listeriosis [55].

The consideration of a preventive QS certification system is important within the context of the explicit obligations placed on food business operators through EU food law to undertake such monitoring both against microbiological criteria in food and, in the case of *L. monocytogenes*, within the food production environment, to validate the effectiveness of their food safety management systems. Official control analyses serve a different purpose, and are required to be risk based, as opposed to representing food production, which generally occurs at much lower frequency. For example, in Austria, 35,000 food samples are annually taken by public authorities by a factor of >5.

## 4. Conclusions

The increasing trend of listeriosis incidence in Austria, from a mean value of 0.17 per 100,000 inhabitants from 2000 to 2005 to a mean value of 0.4 from 2009 to 2018 (Austrian Agency for Health and Food Safety—AGES, 2018; https://www.ages.at/download/0/0/c38f0d95e095fe7e74162ddae9052a4c532450db/fileadmin/AGES2015/Themen/Krankheitserreger_Dateien/Zoonosen/Zoonosenbroschuere_2018_1o_Din-A4_BF.pdf; accessed on 9 June 2021), emphasizes the requirements for effective strategies that meet the control needs of the national public health system and food manufacturers. The liquid-based sampling strategy within a *Listeria* monitoring program at the plant level appears to be superior to solid surface monitoring. Cheese smear liquids seem to have good utility as an index of the contamination of cheese up to that point in production. Multiple volumes of liquid phase, as implemented with our semi-quantitative approach, seem to improve the likelihood of detection, which is consistent with improved epidemiological sensitivity. Monitoring results show a downward trend in *Listeria* prevalence within this matrix, at least for industrial cheese production, which is thereby consistent with improved hygiene in cheese processing environments and cheese products. Modeling or performance testing of this new semi-quantitative approach against the ISO method would be important to more concretely assess the potential for *Listeria* minimization in cheese production.

## Figures and Tables

**Figure 1 foods-10-01977-f001:**
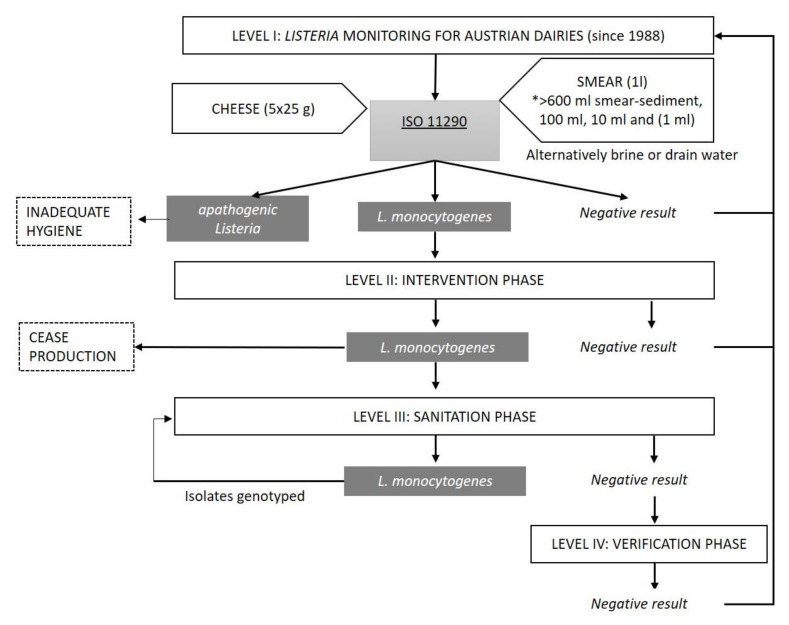
Flow chart displaying the structure of the Austrian *Listeria* monitoring and intervention program. Abbreviations: *, semi-quantitative liquid-based sample quantities.

**Table 1 foods-10-01977-t001:** Numbers of small and industrial food establishments (FBOs) that tested positive for *L. monocytogenes* and other *Listeria* spp., which participated in the *Listeria* monitoring program (2009–2018).

	Small FBOs (HBLFA)		Industrial FBOs (IFFV)	
Year	*L. monocytogenes* Positive/Total *n* (%)	*Other listeria* spp. Positive/Total *n* (%)	*L. monocytogenes* Positive/Total *n* (%)	*Other listeria* spp. Positive/Total *n* (%)
2009	6/51 (11.8%)	7/51 (13.7%)	2/8 (25.0%)	2/8 (25.0%)
2010	8/64 (12.5%)	10/64 (15.6%)	1/9 (11.1%)	2/9 (22.2%)
2011	3/56 (5.4%)	8/56 (14.3%)	2/9 (22.2%)	3/9 (33.3%)
2012	2/63 (3.2%)	13/63 (20.6%)	0/9 (0%)	4/9 (44.4%)
2013	2/68 (0.3%)	11/68 (16.2%)	3/7 (42.9%)	3/9 (42.9%)
2014	2/73 (2.7%)	14/73 (19.2%)	2/8 (25.0%)	5/8 (62.5%)
2015	0/75 (0%)	13/75 (17.3%)	1/7 (14.3%)	2/7 (28.6%)
2016	2/74 (2.7%)	6/74 (8.1%)	2/7 (28.6%)	2/7 (28.6%)
2017	3/74 (4.1%)	12/74 (16.2%)	2/6 (33.3%)	2/6 (33.3%)
2018	2/75 (2.7%)	11/75 (14.7%)	1/7 (14.3%)	2/7 (28.6%)
Mean	3/67.3 (4.5%)	10.5/67.3 (15.6%)	1.6/7.7 (20.8%)	2/7.7 (28.6%)

Abbreviations: FBOs, food business operations supervised by Higher Federal Teaching and Research Institute Tyrol (HBLFA) and Institute of Food Safety, Food Technology and Veterinary Public Health (IFFV); *Listeria* spp., *Listeria* species other than *L. monocytogenes* differentiated by *iap* PCR [32].

**Table 2 foods-10-01977-t002:** The number of smear liquid samples tested and the rate of *L. monocytogenes* and other *Listeria* spp.-positive results found.

Year	Small Dairys (Western Austria; HBLFA)					Industrial Cheesemakers (IFFV)				
	*n*	*L. monocytogenes*	(%)	Other *Listeria* spp.	(%)	*n*	*L. monocytogenes*	(%)	Other *Listeria* spp.	(%)
2009	475	19	4	13	2.7	189	5	2.1	13	6.9
2010	620	27	4.4	12	1.9	503	3	0.6	68	13.5
2011	394	3	0.8	10	2.5	881	12	1.4	27	3.1
2012	441	2	0.5	23	5.2	774	0	0.0	70	9.0
2013	441	3	0.7	21	4.8	711	3	0.4	22	3.1
2014	516	2	0.4	22	4.3	702	2	0.3	19	2.7
2015	523	0	0.0	21	4.0	1535	24	1.6	14	0.9
2016	512	3	0.6	9	1.8	634	8	1.3	14	2.2
2017	544	3	0.6	24	4.4	752	45	6.0	46	6.1
2018	577	5	0.9	14	2.4	513	9	1.8	51	9.9
Total	5043	67	1.29	169	3.4	7194	111	1.55	344	5.74

Abbreviations: Small dairys and industrial cheesemakers supervised by Higher Federal Teaching and Research Institute Tyrol (HBLFA) and Institute of Food Safety, Food Technology and Veterinary Public Health (IFFV); *Listeria* spp., *Listeria* species other than *L. monocytogenes* differentiated by *iap* PCR [32].

## Data Availability

Not applicable.

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
