# Peer review of "Monitoring by a Sensitive Liquid-Based Sampling Strategy Reveals a Considerable Reduction of Listeria monocytogenes in Smeared Cheese Production over 10 Years of Testing in Austria"

_foods, 2021, doi:10.3390/foods10091977_

Round 1

Reviewer 1 Report

The article is greatly improved with the changes presented in the present form. However, the authors should include a clearer description of the findings they are going to present in the text at the end of their introduction and the objective of this work.

Author Response

Dear reviewer, thank you again for the great contribution to improve the manuscript.

At the end of the introduction the aim of this study was highlighted in green.

"Therefore, the aim of this study was to present the alternative semi-quantitative liquid based sampling strategy to increase the epidemiology sensitivity for the detection of L. monocytogens and other Listeria species. For this purpose, the alternative method was implemented within the framework of Listeria monitoring, at both industrial cheese makers and small-scale dairies located in the mountainous region of Western Austria. By using this approach more than 12,000 samples were tested during 2009 to 2018."

Reviewer 2 Report

Grateful for your active and detailed consideration of comments.

For this reviewer perspective the presentation foes justice to a novel, comprehensive, ambitious and informative project.

Author Response

Dear reviewer, thank you again for the great contribution to improve the manuscript.

This manuscript is a resubmission of an earlier submission. The following is a list of the peer review reports and author responses from that submission.

Round 1

Reviewer 1 Report

The authors present an interesting article about the monitoring of Listeria spp in cheese factories. The article, however, is a little confusing and needs some improvements.

  1. No references should be cited in the abstract. Would you please delete it from this section?
  2. The introduction is adequate, but the objectives of this work should be stated more clearly: the authors will present their findings of the last 10 years of monitorization and its impact on this industry or their analytical methodology while applying it to several samples?
  3. In section 2.1 Materials, the authors do not present any materials, reagents or samples. Therefore, they should rewrite this section.
  4. 2.2 Companies is not clear. The authors seem to describe the cheese industry in Austria, but how many of these companies participated in this monitoring study? How many times per year and with how many samples each time?
  5. The information presented in Table 1 are the results obtained? In this case, this Table should be presented in the Discussion section.
  6. The discussion is interesting but a little confusing. For example, the authors repeat the same words in the same sentences more than once. Examples: only in Line 252 and samples in Lines 252 to 255. Please rewrite the discussion considering this and also according to the objectives clearly stated in the Introduction. 
  7. The Conclusion section is incomplete. The authors could include the major findings and future work. 

Reviewer 2 Report

Overall

The overall opinion of this reviewer is that the quality of work particularly the relatively novel and clearly effective sampling approach (both liquid and multi-volume) and the results of a large-scale chronologically-broad application of that monitoring.

However, the quality of presentation in the draft is a poor reflection of the quality of the underlying work. There is a real lack of scientific and linguistic discipline in the presentation of the findings and the derivation of conclusions. It should be possible to re-formulate the presentation with more attention to what was done, what was found and what that means. Extensive comments provided below for consideration

lines 1-5. The paper is not about mitigation, (there are no details of any mitigation) it is about monitoring strategy and results of that strategy.

Abstract

Line 19 Dairy is unclear term, is that dairy farm, or cheese manufacturer. What is ‘most’?  Is that most of the quantity of dairies or most of the quantity of cheese. Is it really milk-borne outbreaks, or perhaps dairy product associated outbreaks. Is this outbreaks of Listeriosis.

Line 22. Sensitive is used here as an absolute term. And it is used to describe a ‘sample preparation’ method’. Should this be a sampling strategy that uses a liquid matrix in order to maximise epidemiological sensitivity.

In line 21 the reader learns of a method based on ‘production associated liquids’ and then in line 29 that becomes ‘smear liquid samples’. Suggest smear-liquid concept introduced early and consistently

Line 23-25,. This sentence lacks meaning

  • Rapid detection refers to the time from the event until the time of detection being short.
  • AN impending contamination period, refers to a time period in the future when contamination might be more likely to occur at any point in time during that period
  • LM transfer from the environment is presumably the contamination of the food?
  • Allows to minimise is not English

Suggestion: A sampling strategy within cheese production which detects environmental contamination before it results in problematic food contamination, has benefits for food safety management.

Line 28-30 and many other parts of the paper

Contamination rate of samples is not in fact correct. If a dairy has high-volume sample (600ml)  positive and a low-volume sample (1ml)  positive from one sampling, then that is 2 positive samples but just 1 contamination event, so it is rate of contamination events  , not sample positivity.

Line 34 suggests the reduction in positivity might be ‘despite’ a ‘large’ increase. There are indeed hypothesis whereby risk of listeria contamination might increase with increased volume of production, but also many hypothesis to the contrary e.g. improved hygiene in industrial high-volume purpose-designed plants, or indeed lower epidemiological sensitivity within sampling of high volume. Suggest reconsideration of wording currently expressing surprise or inconsistency between higher volume and lower positivity.

Line 35-37

This is not a conclusion in the conclusion section 4 line 313. It is also not a conclusion in any way supported by the data presented. There are indeed data to show that food safety has improved in industrial cheese producers. No data were generated or presented on increased awareness or improved hygiene measures, and no data were shown to attribute food safety improvement to that increased awareness or to improved hygiene measures.

Introduction

Line 43. Highly contaminated? Is that high prevalence or with high quatative load, or both. Suggest more clarity, as the present study is semi-quantitative.

Line 45 the reference to CDCD is given as an URL, compared to line 317 ‘anonymous’ references’ Consistency neded

Line 46 begins with ‘currently’ and then refers to a time period which hasn’t yet ended 2020-2021, and then refers to a 2017 reference. This is unclear. What time period. What does contaminated mean is it exceeding 100cpg? Need to explain EU RASFF firstly as an EU tool, secondly only for cases of trade between EU MS, and not for contamination within one MS< and finally not an epidemiological tool merely reflecting instances when detections are reported. There is no denominator for these 90 notifications. It shows nothing other than the fact that LM may be detected if someone looks when food is traded amongst EU MS.

Line 49. This sentence is unclear. What is a globally relevant LM? What LM is not globally relevant? Genoptypes were identified by subtyping? Suggest sentence along the liens of. Significant genetic diversity has been identified within LM genus with molecular epidemiology methods.

Line 53. Some colleagues, perhaps ‘other research groups’. Affinity suggests causation as opposed to association, so is incorrect. Harbouring suggests making active effort to protect within their structure, when it is in fact simply being hypervirulent. Language might be either harbouring genetic determinants of hyper virulence or actually being hypervirulent strains. Unclear in wording what affinity is described , is it lineage I affinity to ‘harbour’ these strains or is it the dairy niche to harbour these lineage strains which happen to be these hypervirulent strains/.

Line 56 Globally relevant. Does this mean ‘outside of Austria’?

Line 62 Major role?  Reconsider language and absoluteness of this statement.

Line 65, sampling concepts do not reduce LM, they detect and quantify LM. Hygiene and decontamination reduce LM not sampling. Suggest direct impact on surveillance to verify effectiveness of LM controls withinfood safety management systems.

Line 67 says most milk undergoes a heating process prior to processing, line 225 contradicts this and says small producers often use raw milk??

Line 68, point is valid and well made, but re-contaminated should be considered as contaminated. And multiple suggests initial then again and again, so  is probably less accurate than saying contaminated at any or all of multiple steps.

Line 70. This focus is a legal obligation of operators in their own checks, regulation 2073/2005.

Line 71. Detection of LM does not necessarily mean the FPE is colonised by LM, it may reflect recent contamiantion.

Line 74, the linguistic nuances of persistence vs colonisation should be explained, if authors wish to use the words to mean different things.

Line 76. Recalls or withdrawals (two different things) only occur when LM is detected on the end product at unacceptably high levels when there is a shelf life study, or detected at any level in instances of no shelf-life study, AND where the product has been placed on the market. Recalls are not therefore necessary when product has not been placed on the market.

Line 77 The important point, that the risk of product recalls or withdrawals is minimised by sampling at this point, at smearing before maturation, to allow the result to be known before placing on the market, is not clearly made. Moreover the disadvantage of this pre-maturation approach, whereby contamination during maturation or packaging or slicing would not be detected, should also be mentioned. Finally, surely it also manages risk to public health in addition to risk of product recalls/withdrawals.

Line 81-82. Sentence is unclear. Suggest: Compared to product-contact surface-sampling using friction-swabs, these liquids are a matrix giving a much broader representation of the contamination status by including both cheese components and contact with surfaces inside production equipment e.g smear robot.

Line 85. English ‘is depending’ suggest depends on.

Lien 86, regularly means at predictable intervals, frequently means at short intervals.

Line 89, make clear that PCR followed culture, not using PPCR as detection method.

Line 85 -93 seems to be about Materials and methods on what sampling strategy weas employed here, so should be in past tense, depended on, was performed

Line 94-100 seems to describe in the present tense events following detection. Difficult to see this anywhere else other than materials and Methods, as ultimately it supports the reduction in prevalence. This was more than a monitoring project, it was a surveillance project where positive results resulted in interventions to reduce positivity. Consider moving to M&M and writing in past tense.

Line 102, suggest new paragraph, completely different topic. High sample volumes is unclear, suggest high-volume samples or samples with initially high volumes of liquid. The semi quantitative concept comes not from testing high-volume samples but from testing both low volume and high volume, and deducing that presence in high-volume but not in low volume means low quantity. Needs to be clearer. Increased sensitivity, should be clarified as increased epidemiological sensitivity due to higher quantity of sample matrix.

Overall the assertion of increased sensitivity is poorly developed and simply not justified in the data. Increased over what? This introduction should do more to develop the theoretical benefit of the sampling approach a. using liquid and b. using both high and low volumes of liquid. It would be entirely reasonable to speculate on this increasing the sensitivity over e.g only low-volume samples, but no such comparison was provided in the data analysis.

Line 104. Transmission is a new concept, adding to earlier concept of contamination, re-contamination, colonisation, persistence. Suggest for here initial contamination of either the environment or the food.

Line 107. Also line 263. Last century, suggest some sort of association with 1900s vs 2000s, as opposed to a time trend. Suggest avoid this language not appropriate.

Line 108 and elsewhere e.g. line 171. ‘Branch’ is unclear to this reviewer. I think authors might best refer to a ‘sector’ of the Austrian dairy industry.  See line 194.

Figure 1.

Monitoring has various definitions, but ultimately means assessing status with no concept of intervention. This flow chart shows both monitoring and interventions.

Stop of delivery is unclear language. Perhaps cease production and if necessary withdraw/recall product.

Materials

Line 117. Developing this point would really help paper as regards novelty. It is not an issue of testing but an issue of sampling. Sampling of a non-homogenous solid product creates real challenges in consistency and representativeness. Listeria contamination is more likely on the surface rind than the dep tissues of a cheese block. Moreover sampling of a batch of individual cheeses has potential for statistical biases unless true randomisation is rigorously adhered to.

Lien 126. Explain Chamber of Commerce, no meaning to non-Austrian readers, or perhaps say according to official Austrian government data.

Line 137-138. Unclear is this 80% of cheese (overall quantity) or 80% of cheese producers?

Methods

Line 242 past tense, correct, but then all following sentences become present tense. Use past tense.

Line 142. Enrichment volumes suggests these were enriched before centrifugation. Need to clarify sample preparation in more detail.

Line 143. Should ‘all’ be ‘low’ ????

Table 1

This has both Materials (operator participation) and results (case rate) maybe move to results.

Line 167 tries to define L spp as L species other than L mono. But L spp inescapably includes L mono. This reviewer finds it inappropriate to use L spp to indicate non-monocytogenes species. To describe such a category, either use ‘other spp’ or ‘non-mono’ but not L spp. See later line 195 where listeria spp is possibly used to mean any listeria including L mono.

Case rate is unclear. What is a case. Is any of the sample volumes positive. If two sample volumes positive is that one case? If there is positivity in two consecutive sampling periods is that one case or two cases?

Results Discussion

Line 171. There sems to be introduced here a whole new part of the study, not described in Material & methods, a survey of technical magers, and a reporting ot the results. The nature fo this survey needs to be described, and the analysis that allowed a general statement about what this population ‘perceived’ on the basis of this survey needs to be provided. What questions were asked, what was response rate, was there a range of responses. What is definition of medium risk what is high level of control. As currently worded this inclusion detracts from the scientific merit of the paper.

Line 172 how can an alleged perception of medium risk and alleged high level of control be regarded as an illusion of invulnerability. In order to extrapolate from the presence of an illusion of invulnerability, readers should be given the basis for that assertion.

Line 178 Recurrent issue: early referencing. There ae are two sentences describing the work of Magdovitz et al. The reference should come after the first sentence referring to the study.

Line 181. Comparable to our concept, appears to refer to the smear-liquid concept. But it actually probably refers to the breadth of data across food producers and batches of cheese. Perhaps explain more clearly which ’concept’. Eu baseline survey deserves mention.

Line 186. Suggest … identified in 19 cheese samples, in the same way as polish study is reported.

Line 192. New concept ‘cross-contamination’, adding to contamination, re-contamination, transmission, colonisation. Need to define, presumably meaning one contaminated batch results in contamination of a different previously non-contaminated batch. Is this correct? Or is it a high risk of contamination?

Line 199. What is cheese paste? This sentence in 198/199 has no clear basis, it was not lookd at in the present study so it needs a reference. It is followed by a sentence saying this (rind>paste) was ‘also’ the outcome of a particular study which mentions two cheese types as most frequently contaminated. The link to rind>paste is unclear.

Line 201. Very unclear what is meant with word mandatory. Perhaps the authors wish to recommend that both smear and rind are always included in the sample matrix. But simply stating something is mandatory is strange unless it is made mandatory e.g in legislation? The linkage to previous paragraph on rind/paste to this statement on rind/smear is unclear.

Line 205. Gone from ‘Case rate’ in Table 1 to ‘Contamination event’ here, and then in line 209 to contamination event rate.  Unclear what is an ‘event ‘e.g. if one plant is positive in two consecutive samplings is that an event. Please clarify. Earlier parts suggest intervention occurs following positivity, does intervention separate events?

Lien 204/205. This sentence is not worded well, average detected at an average. The average events is the mean number of such an events within the study population in any one year of the study. Suggest average of 10.5 non-mono Listeria events were detected annually participating operators, the number of which averaged 67 and ranged from 51-75.

Listeri spp being defined as other (non-mono) listeria is very confusing for this reviewer. Please reassess this terminology. If staying with this, then suggest that it is made clear here for readers that 10.5 excludes L mono events. Line 210 refers to ‘the genus listeria spp’, is that a changed terminology from that point forward??

Line 210 move from events to proportion of samples positive, so one event with two different positive voluems now becomes tow positive data points? This is confusing and unclear?

Line 213 states that a higher proportion of L spp contamination events was expected. Which version of L spp is this other/non-mono or aall including L mono?? Higher than what? Proportion of what? What is the basis for this expectation? Expected by whom?

This expectation is then states as ‘shows’ that L spp contamination pinpoints something. No data is provided for this assertion. No results show higher likelihood of non-mono before a L mono contamination event, either in time or in operators. Even if that were shown that would support but not show that the issue was ‘hygiene deficit’ perhaps instead raw material load?

Line 217. Location of testing is irrelevant, suggest higher in small producers than in industrial-scale producers. Need statistical analysis to substantiate the ‘substantially lower’. Is this difference statistically significant?

Line 218, Jumping from non-mono to L mono, is confusing for readers. Is prevalence in smears the same as contamination events (tow different positive sample volumes = 1 event and 1 positive smear) and different to the prevalence in smear samples?

Line 223 explain Listeria spp as listeria other/non-mono if that is the case?

Line 226 says prevalence (of something?) similar in industrialised and small, whereas line 216 says non-mono lower in small than industrialised. Suggest lie 226 clarified to refer to L mono.

Line 229 contaminated due to recontamination. Suggest simply becomes contaminated after coagulation from L mono within the production environment, as opposed to contamination persisting from the raw material through the coagulation process.

Line 225-230. Lots of other hypothesis are also valid. E.g. two different contamination pathways for small vs industrialised. Lower non-mono prevalence could be developed more in the stated hypothesis.

Line 236. Line 256.Some modelling or microbial validation of this hypothesis would be interesting, or as minimum some reference to the ISO method having higher likelihood of detecting Listeria when present in higher quantities.

Line 238. Not sensitive enough is correct English. But scientifically the issues is epidemiological sensitivity.

Line 239 contradicts lines 235-238 by suggesting higher volumes are not good.

Line 240-242 should be framed as hypotheses to explain. It is not correct to say we explain this by a, b and c. More correct to say potential explanations might be a, b or c.

Line 249 Sentence not correct English. Suggest .. not sufficient to detect a contamination event, therefore the more expensive approach of testing more than one sampling volume should be incorporated.

Line 252 253 and 254.. Line 252 seems to talk about the proportion of all positive results, whereas lins 2253 and 254 seem to talk about the positivity rate of samples. This is confusing. Similarly line 377 is about percentage positivity, whereas line 260 is about percentage of positive results. Suggest consistency of one parameter to describe relationship between sample volume and positivity.

Line 255 Needs to be past tense.

Line 256 see comment on line 236.

Line 262, Suggest refer to some data to support the ‘clear trend’. No pre-200 data are shown in either table 1 or table 2???? some statistical analysis would be useful. Is the decrease significant?. Is this Trend about non-mono Listeria spp or Listeria monocytogenes. If the pre-200o data are only available for industrialised cheese, is this trend only demonstrated for that sector? If so then the sentence should clarify this. Please show data to support this.

Line 267. What is high? Need a reference and or a figure/range.

Line 270. 'in spite of...' The assumption that restructuring should be expected to increase Listeria prevalence in unreasonable. It is not unreasonable to believe that restructuring for greater production by smaller number of operators, might reasonably include infrastructure to improve hygiene e.g clean-area barriers and equipment to minimise manual handling and designed to facilitate cleaning. Also Darwinian selection away from less survival of poor operators. If the authors wish to make an assertion of restructuring to less producers and more volume being synonymous with increased listeria risk, then some references should be provided, as opposed to having it as an axiomatic assumption.

Line 276. Sentence needs to be developed. Monitoring is a prerequisite for detection is axiomatic. The point seems to be that frequent monitoring aids early detection, and prevents contamination and placing contaminated food on market.

Line 282. Poor English. Rephrase. At least once

Line 282, 284. Need some discipline between perhaps initial once-off contamination of FPE, ‘introduction’ vs contamination fo food

Line 2287 288. Consider rephrasing. Hygienic pressure too to prevent them surviving. Consider hygienic pressure not high enough to prevent them surviving or too low thereby allowing them to survive.

Line 289. This is a conclusion from earlier discussion but not correct according to earlier discussion. The earlier finding was that high volume sampling alone was not the answer, but in fact multiple samples with both high and low volume as optimal.

Line 293. Sampling Bias is not entirely correct. It is more about a challenge of a challenge in maintaining sample representativeness and consistency/comparability.

Lien 298 Preventing foodborne hazards is simply not reliant on sampling. It is supported by sampling It is reliant on hygiene measures, and sampling can verify the effectiveness of those measures.

Lines 298 -300 should come earlier, perhaps in introduction.

Line 306. Refers to. Suggest ‘is consistent with’

Line 308. Revealed a push by, is incorrect English. Perhaps… in part contributed to by a particular high-profile cheese-borne outbreak of Listeriosis.

Line 309-311. This is not a valid comparison, and is indeed completely consistent with Eu food law. It would be much more accurate to say …. is important within the explicit obligations on Food business operators through EU food law to undertake such monitoring both against microbiological criteria in food and in the case of L. monocytogenes within the food production environment, to validate the effectiveness of their food safety management systems. Official control analyses serve a different purpose, and are required to be risk based, as opposed to representing food production,  generally occurring at much lower frequency. For example in Austria...

Conclusions

Line 313 Acceptance? This is not a conclusion of this study No data are presented to support this clnclusion

Line 313 Incidence in people. This is not a conclusion of this study? Much more appropriate as part of introduction.

This reviewer's conclusions

  • Liquid-monitoring looks better than solid -surface monitoring.
  • Cheese smear liquid seems to have good utility in as an index of the contamination of cheese up to that point in production
  • Multiple volumes of liquid phase seem to improve likelihood of detection, consistent with improved epidemiological sensitivity.
  • Multiple different volumes seem to give useful semi-quantitative information.
  • Monitoring results are show downward trend in Listeria prevalence this matrix, at least for industrial production thereby consistent with  improved hygiene in these environments and in this product.